# Hock Lesions in Dairy Cows in Cubicle Housing Systems in Germany: Prevalence and Risk Factors

**DOI:** 10.3390/ani13182919

**Published:** 2023-09-14

**Authors:** Cindy Freigang, Katharina Charlotte Jensen, Amely Campe, Melanie Feist, Andreas Öhm, Marcus Klawitter, Annegret Stock, Martina Hoedemaker

**Affiliations:** 1Clinic for Cattle, University of Veterinary Medicine Hannover, Foundation, 30173 Hannover, Germany; cindy.freigang@tiho-hannover.de (C.F.); martina.hoedemaker@tiho-hannover.de (M.H.); 2Department of Biometry, Epidemiology and Information Processing (IBEI), WHO Collaborating Centre for Research and Training for Health at the Human-Animal-Environment Interface, University of Veterinary Medicine Hannover, Foundation, 30173 Hannover, Germany; 3Clinic for Ruminants with Ambulatory and Herd Health Services, Ludwig-Maximilians Universität Munich, 80539 Munich, Germany; 4Clinic for Ruminants and Swine, Faculty of Veterinary Medicine, Freie Universität Berlin, 14163 Berlin, Germany

**Keywords:** hock lesions, housing, cubicles, bedding, welfare, dairy, rubber mats, litter

## Abstract

**Simple Summary:**

Cows often have hairless patches, wounds, or swellings at the hocks. These hock lesions are indicators of suboptimal housing conditions. This study assessed how many cows in Germany had hock lesions. Moreover, we analysed which factors were associated with the occurrence of hock lesions (risk factors). Therefore, 554 dairy farms in three different regions in Germany were visited and the hocks of 66,681 cows were assessed. Only cows kept in cubicle housing systems, where cows can move freely, were included. Between 66% and 80% of the cows had at least one hairless patch on the hocks. Moreover, between 8% and 14% of cows had at least one wound and/or swelling at the hocks. Cows that were kept in pens with cubicles that were deep bedded had a lower chance of hock lesions than cows that were kept in cubicles with rubber mats or comfort mats—even if these had a small amount of litter. Cows that were lame, were more likely to have hock lesions. Finally, with increasing age and with a decreasing body condition, the chance of hock lesions rose.

**Abstract:**

Hock lesions in dairy cows are an important indicator of animal welfare, in particular housing conditions. The aim of this study was to assess the prevalence of hock lesions in dairy cows kept in cubicle housing systems in three structurally different regions of Germany and to derive recommendations from risk factor analyses. Lactating and dry cows kept in cubicle housing systems were assessed for hock lesions (north: 206 farms with 20,792 cows; south: 156 farms with 8050 cows; east: 192 farms with 37,839 cows). Risk factor analyses were conducted using multi-factorial logistic regression models. The median prevalence of hock lesions (hairless patches, wounds, and/or swelling) at farm level was 79.8% (SD: 25.0; north), 66.2% (SD: 31.0; south), and 78.5% (SD: 26.3; east). The mean prevalence of severe hock lesions (wounds and/or swelling) at farm level was 12.5% (SD: 11.3; north), 8.0% (SD: 13.5; south), and 14.4% (SD: 17.9; east). Cows kept in pens with rubber mats or mattresses (with or without a small amount of litter) had a particularly higher chance of hock lesions compared with cows kept in pens with deep-bedded cubicles (OR: north: 3.1 [2.3–4.2]; south: 8.7 [5.9–13.0], east: 2.0 [1.7–2.4]). The study showed that hock lesions are a widespread problem on German dairy farms with cubicle housing systems. Deep-bedded cubicles are likely to reduce hock lesions and increase cows’ comfort.

## 1. Introduction

Nowadays, increased emphasis is placed on animal welfare in our society [1,2]. Welfare is defined as a state of physical health and psychological well-being of the animal within itself and with its environment. However, temporary adverse environmental conditions and adaptation efforts do not necessarily affect animal welfare as long as the animal has experienced that it is able to cope with the demands and the current situation [3]. However, lesions, such as swellings or wounds, indicate that an animal’s adaptation strategies have been exceeded and animal welfare is compromised [3]. Thus, lesions are an important indicator of housing conditions and also of the welfare of livestock [3,4].

A lesion is defined as damage, injury, or disturbance of an anatomical structure or physiological function [5]. Lesions of the hocks in terms of hairless patches, wounds, or swellings are the most common lesions in dairy cows [6,7,8]. Internationally, multiple studies have investigated the prevalence of hock lesions. For example, Zurbrigg et al. [9] investigated the prevalence of hock lesions in tethered stalls. In the following years, further studies were carried out that assessed hock lesions in free stalls. In Canada and Norway, the prevalence of hock lesions (hair loss, swellings, and wounds) was found to be relatively high at 61% [6] and 62% [8]. The methodologies between these studies vary and therefore, results have to be compared with caution.

Looking at the current management of dairy cows in Germany, more than 80% of the farms keep their cows in cubicle housing systems in 2020 [10]. It can be assumed that the number of animals in cubicle housing systems will continue to increase, as tethering, which is still practised especially in the south of Germany, is declining [11] due to regulations but also due to the high amount of work involved. Cubicles are intended to enable cows to lie down in a manner typical of their species. At the same time, they should prevent cows from lying on their excreta and thus reduce the risk of mastitis. Cows lie down for about 12–14 h a day [12]. During this time, udder circulation is promoted, ruminating activity is increased, the limbs are relieved, the claws can dry off, and the corium in the claw is supplied with more blood. Thus, a high lying time has a positive effect on the productivity of cows, but is also an indicator of welfare [12].

A mean prevalence of hock lesions of 21.8% was found in dairy herds in northern Germany in 2015 [13]. It can be assumed that a similar prevalence exists nationwide. However, comparable studies in other regions of Germany are still lacking. In Germany, dairy cow farming is characterised by major structural differences [14]. For example, in the federal states that belonged to the former German Democratic Republic until reunification, large farms can be found, which in most cases employ staff from foreign countries [10]. In northern Germany, there are mostly family-run farms with an average of about 112 cows [10]. In southern Germany, on the other hand, the average herd size is only 52 cows. Here, numerous farms are still run on a side-line basis. It is conceivable that not only the management and husbandry condition differ in these regions but also the prevalence of hock lesions.

The aim of this study was, therefore, to determine the prevalence of hock lesions for these three structurally different regions in Germany, in order to draw conclusions about the lying comfort and well-being of dairy cows in cubicle housing systems. In addition, risk factors for hock lesions (HL; defined as hair loss, wounds, and/or swelling) and severe hock lesions (SHL; defined as wounds and/or swelling) should be identified in order to generate recommendations for actions helping to prevent the occurrence of hock lesions.

## 2. Materials and Methods

The data evaluated in this manuscript were collected within the study: “Animal health, hygiene and biosecurity in German dairy farms—a prevalence study (PraeRi)” [15,16] which was conducted between 2016 and 2020.

As there are significant structural differences in dairy cow husbandry in Germany [14], three regions were regarded separately: in the northern region, farms in the federal states of Lower Saxony and Schleswig–Holstein; in the eastern region in the federal states of Mecklenburg–Western Pomerania, Brandenburg, and Thuringia; and in the southern region, farms in Bavaria were investigated. In order to ensure representativeness regarding the sizes of the farms, three farm size classes were introduced based on the data of the Dairy Herd Improvement Services (DHI) or the Association for Milk Testing (Bavaria), defined differently depending on the region (Table 1). About the same number of small, medium, and large farms were visited in each region.

Randomly selected farms from the Animal Origin Assurance and Information System (Herkunftssicherungs- und Informationssystem für Tiere) ) in the north and east regions and the Association for Milk Testing in the southern region were contacted by mail and asked to participate voluntarily. All farms were visited once by trained study veterinarians. The training was conducted in a meeting before the farm visits, using photographs. Scoring charts were used; the inter-observer reliability between all researchers collecting the data was assessed three times before (meeting 1) and during (meeting 2 and 3) the study period. Each of these assessments took place in the form of a practical course on a farm. During the first assessment, 15 cows were scored, 48 cows were scored during the second assessment, and 59 cows were scored on the third assessment date.

For the purposes of this study, only farms that kept their cows predominantly (85% or more) in loose housing with cubicles were evaluated. Thus, farms where more than 15% of the cows were on pasture, tethered, or in other housing systems at the date of the farm visit were excluded. In addition, within the remaining farms, animals were excluded if they were not kept in free stalls.

Lactating and dry cows were scored regarding locomotion, body condition, lesions on the hocks, neck, and back, alteration of the tail, and hygiene. The scoring system by Kielland et al. [6] was modified to assess the lesions on the hocks. This scoring scheme does not provide information how to handle dirty hocks and how hocks with swelling and wounds should be assessed. Therefore, we changed score 5 and added an additional score 6. In principle, both hocks of the cow were considered, but only the more severe higher score was assessed (Figure 1). We decided to consider the worst hock, as the main goal of our study was to identify risk factors. If we had assessed only one hock, this probably would have caused an underestimation of the effect of the risk factors. If one hock was too dirty to be evaluated, the other hock was assessed.

The hock was regarded from behind and laterally, according to the following scale:

1 = no change

2 = hairless area or clear hair breakage (skin visible), no wound or swelling

3 = swelling (obvious increase in size) without wound, with or without hairlessness

4 = wound (scabs or transection of the skin) without swelling, with or without hairlessness

5 = swelling and wound, with or without hairlessness

6 = no score due to high degree of soiling of both hocks

7 = both hocks not assessable due to other reasons

For reasons of practicability, the number of cows to be scored and cubicles to be measured were limited based on a sample size calculation. The sample size for the number of cows is shown in Table 2. The sample calculation of the scoring is based on a confidence interval of 95%, an expected prevalence of 40%, and a precision of ±5%. The number of dry and lactating cows was stated by the farmers. Then, the percentage of cows to be scored was calculated. If, for instance, every third cow was to be scored, one cow was scored and marked and the two cows next to the observer were not scored but were marked. This scheme was the same in all pens with cows to achieve an equal distribution of cows.

For the measurement of cubicles, the sample size calculation was based on a confidence interval of 95%, a standard deviation of 10, a precision of ±5%, and a farm size of 10–1500 cubicles. For both sample size calculations, NCSS PASS version 13.0.8 was used.

The number of cubicles to be measured was based on the total number of cubicles on the farm and was the same for all regions: for up to 29 cubicles, 10 cubicles were measured; if there were 30–49 cubicles, 15 cubicles were measured; if there were 50–99 cubicles, 17 cubicles were measured; and if there were 100 or more cubicles, 18 cubicles were measured. To ensure that the cubicles were measured evenly across all compartments, the number of cubicles to be measured was divided by the total number of cubicles for dry or lactating cows. Then, for example, every fifth or twentieth cubicle was measured during a walk through all the compartments.

The risk factors were chosen based on the hypotheses stated before the farm visits. The data were collected using standardised forms and questionnaires on the day of the farm visit or were retrieved from the DHI testing and herd registers.

The compartments with dry and lactating cows were assessed regarding hygiene, type of cubicles, and stocking rate. In addition, the farmer was asked about farm management practices. Table 3 gives an overview of the risk factors, their source, definition, and, if applicable, categories.

All data was transferred to an SQL database and checked for plausibility where possible. The statistical analysis was conducted using SAS EPG (version 7.1) or SAS (version 9.3). All analyses were carried out for each region separately. For the first hypothesis, the description of the prevalences, analyses were carried out at farm and at cow level.

For the risk factor analyses, the outcome (hock lesions) was dichotomised in two different ways: on the one hand, it was considered which factors had an influence on the occurrence of all types of lesions (HLs, score 2–5, hairless area, wound and/or swelling). Secondly, the risk factors associated with the occurrence of severe hock lesions (SHLs, score 3–5, wound and/or swelling) were assessed. A descriptive analysis of the risk factors in relation to the two target variables was carried out. The categorisation was adjusted, if necessary. For example, only a few farms in the southern region were in the transition period from conventional to organic farming. Therefore, the farms in transition were added to the organic farms. The association between the risk factors was determined using Spearman’s correlation coefficient (quantitative × quantitative variable), the Kruskal–Wallis test (quantitative x qualitative variable), or Cramér’s V (qualitative x qualitative variable). Among the variables, only bedding and type of cubicle showed a medium to moderate association (Cramér’s V: north = 0.21, south = 0.45, east = 0.48). Obviously, there is a logical connection between these two factors as deep-bedded cubicles always have litter. Only cubicles with rubber mats and those with comfort mats vary with regard to the occurrence of litter. We considered combining cubicle type and litter into one factor but then decided to regard both variables in the multifactorial modelling to determine the effect of litter itself. For the modelling, all cows for which no complete dataset was available were removed in order to keep the quality of the models comparable. Seven risk factors (grazing time, locomotion, bedding, type of cubicle, organic or conventional management, body condition, and age) were regarded as mandatory as these were either confounders or were considered indispensable. The other (potential) risk factors (cubicle dimensions, soiling, cow–cubicle ratio, lactation stage) were only included in the multifactorial model if they had a *p* value > 0.15 in the single factorial analyses. The construction of the multi-factorial model was carried out at cow level including the farm as a random effect in several steps. First, all risk factors and confounders were combined. Then, the potential risk factors were selected backwards. The variables with the largest *p* values were removed step by step until only variables with *p* < 0.15 remained in the models. This modelling procedure was carried out for all three regions (north, south, and east) and both outcomes.

## 3. Results

### 3.1. Study Population

Figure 2 displays the number of farms and cows included or excluded for descriptive and inductive analyses. The proportion of the target population that was analysed is shown in Table 4. Only 9% (north and east) and 6% (south) of the contacted farms accepted the invitation for participation.

The agreement of the observers on the first meeting was weak (weighted kappa: 0.44–0.52) but improved to moderate (meeting 2: weighted kappa: 0.54–0.65; meeting 3: 0.59–0.69) [26,27]. No statistically significant differences were observed between the regional teams.

**Table 4 animals-13-02919-t004:** Comparison of cows and farms included in the analyses with the target population [28].

**Region**	**Cows in Target Population** [28]	**Cows in Descriptive Analyses**	**Proportion**	**Cows in Inductive Analyses**	**Proportion**
North	1,261,108	20,792	1.6%	17,332	1.4%
South	1,208,640	8050	0.7%	6451	0.5%
East	574,789	37,839	6.6%	32,712	5.7%
	**Farms in Target Population** [28]	**Farms in Descriptive Analyses**	**Proportion**	**Farms in Inductive Analyses**	**Proportion**
North	14,260	206	1.4%	196	1.4%
South	32,564	156	0.5%	139	0.4%
East	2256	192	8.5%	183	8.1%

### 3.2. Prevalence of Hock Lesions

At farm level, 66–80% of the cows had hock lesions (HLs) on average (median; Table 5). On the other hand, the median number of cows per farm with no changes was 18–32% (Table 5). The prevalence of severe lesions (SHLs) averaged over the herd was 8–14% (Table 5), depending on the region.

The prevalence of HLs at cow level ranged from 62 to 74% (north 73.8% [14,156/19,658]; south 61.6% [4543/7.372]; east 71.6% [26,360/36,807]). The prevalence of SHLs ranged from 12 to 20% (north 14.4% [2836/19,658]; south 12.6% [926/7372]; east 20.7% [7610/36,807]). In all three regions, 3–8% of cows´ hocks (north 5.5% [1134/20,792]; south 8.4% [678/8050]; east 2.7% [1032/37,839]) could not be assessed. The reasons for this were mainly heavy soiling, but also included when the cows ran away, or it was too dark in the barn. A detailed overview is shown in Figure 3.

### 3.3. Risk Factors for Hock Lesions

Two approaches were used for the risk factor analyses of hock lesions in dairy cows in cubicle housing systems. First, the factors contributing to the occurrence of hock lesions were investigated. This was divided into “no change” and “at least one hock with a hairless area and/or swelling or wound” (HL). Secondly, it was investigated which factors caused severe changes. For this purpose, the classification was made into no change or hairless area and wound and/or swelling (SHV) on at least one hock.

Appendix A show the results of the descriptive analysis for HLs and SHLs. Table 6 and Table 7 show the results of the multi-factorial analyses for HLs and SHLs.

Regarding the type of cubicle, the fewest SHLs were found in cows from pens with deep-bedded cubicles and the most SHLs in cows from pens with rubber mats or comfort mattresses. These results are illustrated in Figure 4. The multi-factorial results showed that cows kept in pens with deep-bedded cubicles had significantly lower chances of HLs and SHLs compared with all other cubicle types (Table 6 and Table 7).

The chances of HLs and SHLs were twice as high for lame cows compared with non-lame cows (Table 6 and Table 7). A clear increase of the prevalence of SHLs could be seen with increasing locomotion scores (Figure 5).

A high BCS was associated with a lower chance of hock lesions in all three regions (Table 6 and Table 7). Regarding management, cows from conventionally managed farms had a two to four times higher chance of hock lesions compared with organic farms (Table 6 and Table 7).

Concerning the soiling of the lying area, the presence of litter, the stocking density, the grazing time, and the lactation stage, the results differed between the three regions. For example, in the eastern and southern regions, animals without access to pasture had more HLs than animals allowed to graze on average four or more hours a day (Appendix A and Table 6). In the northern region, this effect could not be observed. Concerning SHLs, in the northern and southern regions, the fewest SHLs occurred when the cows had no access to pasture (Appendix A and Table 7). However, in the multifactorial model of northern region, cows without access to pasture had an increased chance of SHLs (Table 7).

Regarding the cubicle dimensions, cows from farms with higher neck bars and wider cubicles had fewer HLs and SHLs (Appendix A). However, in the multifactorial modelling, only a wider cubicle had an effect on the occurrence of HLs. This association was only significant in the northern region. Other effects were not observed (Table 6 and Table 7). As the dimensions of the cubicles did not show any association with the occurrence of SHLs, these factors—as well as the soiling of the lying surface—were not included in the multifactorial modelling.

## 4. Discussion

The aims of this study were to assess the prevalence of hock lesions and to identify risk factors in three structurally different regions of Germany. The results show that hock lesions are a common disorder in dairy cows in Germany. In addition, the cubicle type in particular is associated with the occurrence of lesions on the hocks.

Farms participated voluntarily in this study. This may have led to a selection bias. It can be assumed that farms with poor husbandry conditions did not participate and thus the prevalence was rather underestimated. On the other hand, it is also possible that farms with a high prevalence participated in order to benefit from the free advice within the framework of the study. The low response rate was presumably caused by different circumstances: farmers were contacted by mail and had to contact the study team themselves. Moreover, some farms were contacted that did not fulfil the inclusion criterion as they did not (any more) produce milk. A non-response analysis revealed that 14 out of 20 farmers from small farms in northern Germany who were invited but did not respond to the invitation did not keep dairy cows (any more). However, the response rate in southern Germany was lowest and here, the farms were selected from the Association for Milk Testing, so all the farms delivered milk and where part of the target population. We do not know why so many farmers did not accept the invitation for participation in the study. In the end, prevalences may have been biased. A directed bias effect on the risk factors is unlikely as the selection process did probably not influence the relationship of risk factor and outcome. However, between 8050 and 37,839 cows from 156 to 206 farms were examined in each region, which represents a large sample size. This made it possible to consider a large number of risk factors in the multi-factorial models. Furthermore, due to the separate analyses of the regions, the results can be compared and the consistency of the risk factors in different management systems could be determined.

The inter-observer reliability indicated moderate agreement even though all observers received training beforehand. This led presumably to an information bias. However, as there were no differences between the regional teams, a directional bias is unlikely.

Between 2.5% and 8% of the cows were not included in the study due to hock soiling. This is a surprising result, because previous studies did not report this problem.

### 4.1. Prevalence of Lesions

With a prevalence of HLs of 62–74% and of SHLs of 13–21% at cow level, hock lesions are a common problem in all three regions. Thus, there is a need for further improvement, as hock lesions are a sign of impaired welfare [25]. A higher prevalence of hock lesions was found in northern and eastern Germany than in southern Germany (Table 5 and Figure 3). For severe lesions, the prevalence in the eastern region was higher than in the other regions (farm level prevalence: 20.7% compared with 14.4% in the north and 12.6% in the south). The reason for this may lie in the different farm structures. For example, the focus on larger herds in the eastern region may not provide the same attention for the individual cow. In the study by Rutherford et al. [20], a correlation between farm size and the prevalence of hock lesions was found. In our study, however, this effect could not be observed within the regions.

Various studies have been carried out worldwide on the occurrence of lesions on the hocks. There is a great variance between the different studies with regard to the selection of farms, animal groups, and methodological approach. A comparison of the results is, therefore, only possible to a limited extent. Kielland et al. [6] found a prevalence of 61% for hock lesions and about 7% for severe hock lesions on farms with loose housing systems in Norway. Brenninkmeyer et al. [18], on the other hand, reported a significantly higher prevalence of severe hock lesions in loose housing farms in Germany and Austria – they reported a mean herd prevalence of 50%. Jewell et al. [7] found a hock lesion prevalence of 39% for cows housed in free stalls. With the high prevalences in this and other studies arises the question of how to prevent hock lesions. For this reason, risk factors were identified.

### 4.2. Type of Cubicle and Presence of Litter

The study confirms that the surface of the cubicle is an important factor associated with the occurrence of hock lesions, as cows from pens with raised cubicles had a two to nine times higher chance of lesions than those from pens with deep-bedded cubicles (Table 6 and Table 7). The biggest difference between these types of cubicles is the lying surface. Cattle prefer a soft, malleable surface to lie on [29]. An unfavourable floor condition in the cubicles, such as conventional rubber mats in cubicles without bedding, leads to more hairless areas, crusts, and wounds or swelling in the area of the hocks [6,8,18]. In the study by van Gastelen et al. [17], cows kept in pens with deep-bedded cubicles with diverse types of bedding had fewer and less severe hock lesions than cows kept in pens with comfort mattresses.

Looking at the influence of the presence of litter on rubber mats on the occurrence of lesions, only a relatively weak association was found, which also was not consistent across all three regions.

If lesions at the hock are to be reduced, deep-bedded cubicles are preferable to raised cubicles with rubber mats or comfort mattresses. The occurrence of lesions cannot be significantly reduced by a small volume of litter in raised cubicles with rubber mats. The conversion of cubicles with mattresses to deep-litter cubicles is recommended for the welfare of cows.

### 4.3. Soiling of the Lying Surface

In this study, an association between the soiling of the lying surface and the occurrence of lesions on the hock could only be proven in the eastern region (see Table 6), whereby only medium but not heavy soiling entailed an increase in risk. This could be related to the type of cubicles, as deep-bedded cubicles may become more heavily soiled, but can absorb urine better. Urine and the urea it contains can cause skin irritation [19]. Soiling of dairy cows is mainly due to lying areas covered with faeces and heavily soiled walkways [30]. In this study, in contrast to others, a proportion of cows could not be assessed due to heavy soiling at the hocks. This might have biased the results.

To ensure clean and healthy hocks in dairy cows, regular and careful care of the stall is essential, as otherwise the skin on the hocks can become softened. Soiling of the skin with faeces also induces irritation of the tissue and favours the development of skin inflammation due to massive bacterial contamination [31,32].

### 4.4. Pasture

In this study, the effect of pasture was slightly different between the regions. In the studies by Rutherford et al. [20] and Barrientos et al. [19], pasture played a crucial role in the prevention of lesions. Even a short grazing period had a positive effect. This could not be observed in this study. The multi-factorial modelling showed only a slight increase in the likelihood of HLs when the cows did not have access to pasture compared with those with access of more than four hours a day on average. It is conceivable that farms that do not let their cows out to pasture, or not so much, invest more in lying comfort. Thus, it might be that grazing only has a protective effect, if a lot of time is spent on pasture and the barn equipment also offers appropriate cow comfort.

### 4.5. Cubicle Dimensions

With regard to cubicle dimensions, the study revealed some surprising results. The descriptive analysis showed that on farms with wider cubicles and higher neck bars, fewer HLs and SHLs were observed. However, in the multifactorial modelling only an association between cubicle width and HLs was observed in the northern region. The results might be weakened, as we used a cut-off for the dimensions and did not take the size of the cows into account. In the study by Gieseke et al. [21], a higher prevalence of severe hock lesions was associated with wider cubicles, and no further cubicle dimensions showed an association with hock lesions. The study by Breninkmeyer [18] showed no association between cubicle width and hock lesions. Therefore, we conclude that the width of the cubicles and the height of the neck bars are not crucial risk factors for hock lesions. However, they influence lying behavior and cleanliness [21].

### 4.6. Management (Organic/Conventional)

This study showed that cows from organic farms had significantly fewer hock lesions than cows from conventional farms. Cows from conventional farms had twice the chance of SHL, even though the multifactorial models already adjusted for cubicle design and grazing time. Thus, it can be assumed that in organic animal husbandry, due to the high animal welfare standards [31], other unobserved factors contribute to the avoidance of hock lesions. According to EU legislation on organic farming, various objectives must be taken into account and requirements met [33]. Among other things, the housing of the animals must be species-appropriate and meet their biological and ethological needs. Year-round free-range husbandry is possible in suitable climatic regions. In addition to access to pastures or open-air runs, the stocking density in the barn should also ensure the animals’ comfort and well-being [33]. These factors as well as other unobserved factors might explain the lower prevalence in organic farms.

### 4.7. Lameness

The study showed that the chance of HL and SHL increased consistently in all three regions as the degree of lameness increased.

If lameness develops due to different factors, the duration of laying bouts increases [34]. Several studies showed that cows with a lameness score of 3 according to Sprecher et al. [23] had more lesions on the hocks [6,8,18]. However, it may not be clear whether the lameness is due to pain from the lesions, or the lesions are due to increased lying down and altered standing up and lying down due to the lameness [20]. Common risk factors affecting cow comfort are also conceivably associated [24]. In addition to regular and professional claw trimming to avoid incorrect loading, both the design and quality of the lying and walking areas and the space available are important influencing factors for lameness [24,35]. The degree of impact of lameness on the development of lesions may also depend on individual animal factors. This can be linked to differences in cow size or weight, as well as social status in the herd (e.g., lower ranking cows [18]).

Preventive measures to reduce lameness as well as early detection and consequent treatment of lame cows are thus measures that probably also reduce the occurrence of hock lesions.

### 4.8. Age and Stage of Lactation

With increasing number of days in milk, the cows showed more hock lesions in the descriptive analyses, whereas a significant association could only be shown for the eastern region in the multifactorial analysis. With increasing age, cows had higher likelihood of HL and SHL in all three regions. In the course of an animal’s life, cows go through several lactations. The older cows become, the longer they are exposed to the prevailing husbandry conditions. It is therefore a logical consequence that—if the housing conditions are not optimal—the chance of lesions increases with age and also lactation stage [8,20]. In addition, the weight of the cattle continues to increase after the first calving, so the pressure on the hocks probably also increases. The aim cannot be to eliminate older animals from the group, but rather to improve the housing conditions so that older animals also suffer fewer lesions.

### 4.9. Body Condition

In agreement with most studies [6,18,20,24], hock injuries were also associated in this study with a low BCS; the higher the BCS, the lower were the odds for HL and SHL (Table 6 and Table 7). Maybe a higher BCS provides better cushioning and protection of the hocks against external trauma. Moreover, cows with a high BCS are less likely to be lame [35] and may rank higher and can choose the best cubicles. Finally, further different health problems like, e.g., hypocalcaemia might promote poor body condition as well as hock lesions as cows lie down for prolonged times.

### 4.10. Stocking Density

Differently than expected, in the northern region, cows in pens with decreasing availability of lying places showed fewer lesions on the hock. In contrast, in the study by Barrientos et al. [19], hock injuries occurred more often when the animal-to-cubicle ratio was too high. It was assumed that cows stand up and lie down more often when fewer cubicles are available, and the load on the joints thus increases considerably. In addition, the number of animals lying down on the walkways increases, especially in the case of weaker or lower-ranking animals.

As the resting behaviour of cows is highly synchronised due to fixed milking and feeding times on farms [30], one cubicle should be available for each cow [36,37]. This provides the opportunity for all cows to lie down undisturbed at the same time.

## 5. Conclusions

In summary, this study has shown that hock lesions are a common disorder in dairy cows in Germany. In addition, deep-bedded cubicles made the occurrence of lesions less probable and are thus an adequate measure to prevent hock lesions. The use of litter in cubicles with rubber mats or the use of comfort mats does not come close to the protective effect of deep-bedded cubicles. In addition, the reduction of lameness, optimisation of cubicle dimensions, access to pasture, and optimisation of body condition might contribute to reducing hock lesions.

## Figures and Tables

**Figure 1 animals-13-02919-f001:**
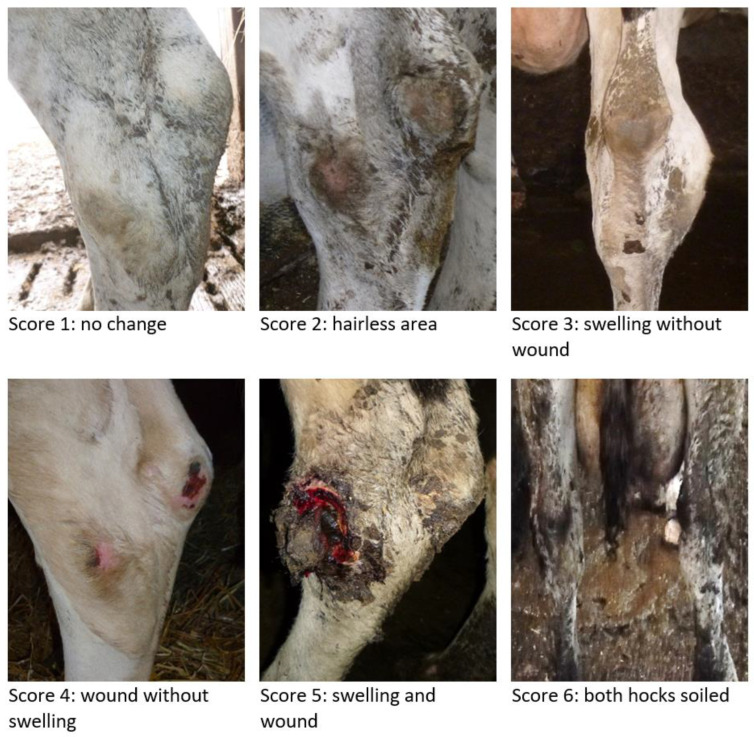
Modified scoring system by Kielland et al. [6] for the assessment of hock lesions in cows in the PraeRi study.

**Figure 2 animals-13-02919-f002:**
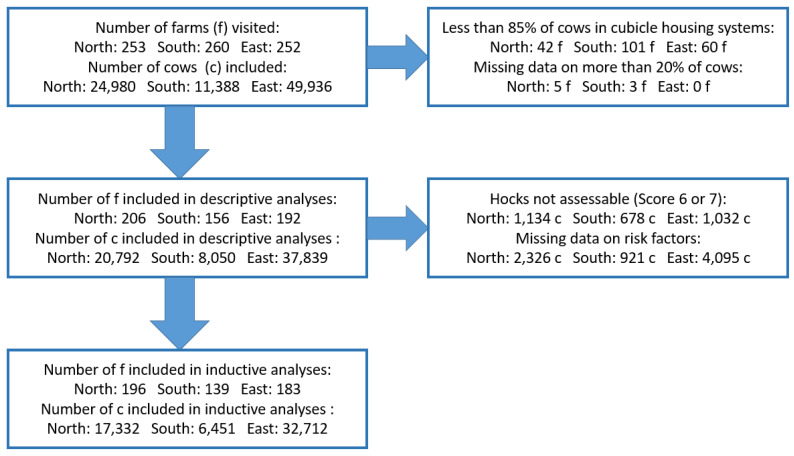
Flow chart demonstrating the number of farms and cows included and excluded due to different reasons in each of the three regions.

**Figure 3 animals-13-02919-f003:**
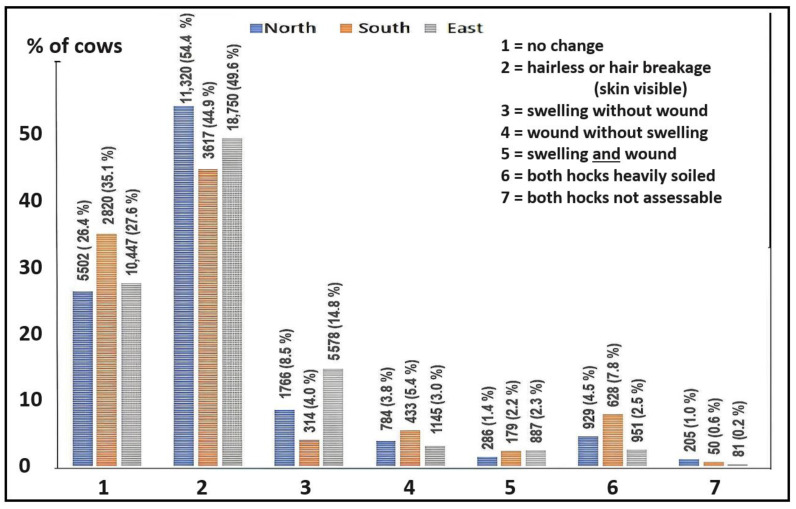
Prevalence of hock lesions on cow level in the PraeRi study.

**Figure 4 animals-13-02919-f004:**
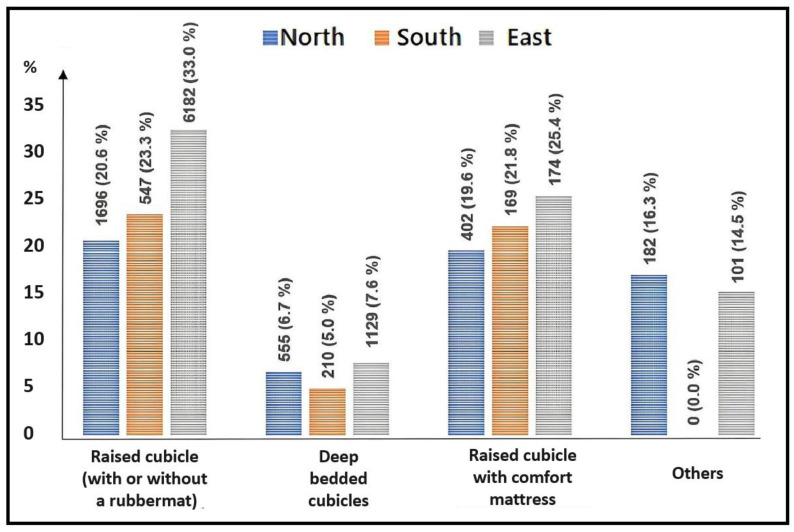
Prevalence of severe hock lesions (SHLs) in dairy cows according to the type of cubicle in the PraeRi study.

**Figure 5 animals-13-02919-f005:**
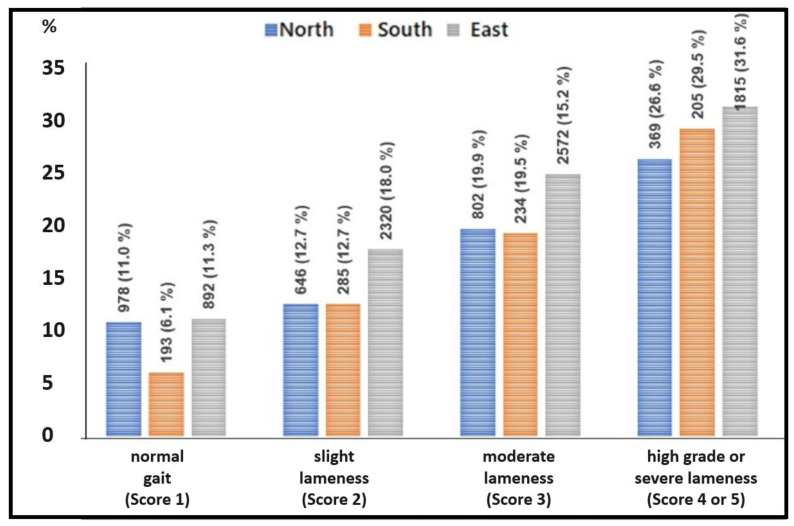
Number and percentage of cows with severe hock lesions (SHLs), stated according to the degree of lameness of cows from cubicle housing systems in the PraeRi study.

**Table 1 animals-13-02919-t001:** Stratification of farm sizes according to the number of cows per regions in the PraeRi study based on data from the Dairy Herd Improvement Services (DHI) or the Association for Milk Testing (Bavaria).

Farm Size	Region		
	North	South	East
Small	<64	<20	<84
Medium	64–116	20–40	84–279
Large	>116	>40	>279

**Table 2 animals-13-02919-t002:** Sampling principle of scoring the cows in the PraeRi study.

Region	Farm Size (Number of Cows per Farm)	Number of Cows Scored
North	<213	All
	>214	213
South	<130	All
	>131	130
East	<165	All
	166–292	166
	>293	292

**Table 3 animals-13-02919-t003:** Overview of the risk factors for (severe) hock lesions in the PraeRi study.

Risk Factor	Source	Level of Assessment	Categories/Definition	Reference
Cubicle type	Assessment by study vets	compartment	Raised cubicles (lying area above the walking surfaces) with or without rubber mats and with no or only little litter Raised cubicles with comfort mattress (flexible rubber mat filled with foam material, e.g.,) with no or only little litter Deep bedded cubicles (lying surface totally covered with litter) Other (all other cubicle types)	[17,18]
Bedding	Assessment by study vets	compartment	Organic bedding material No litter or only lime available	[6,17,18]
Hygiene of the lying surface	Assessment by study vets	compartment	Soiling of the rear half of the cubicle with faeces. Categories: Mainly clean Less than 50% of the lying surface soiled More than 50% of the lying surface soiled	[19]
Animal–cubicle ratio	Assessment by study vets	compartment	Number of animals divided by number of cubicles in the compartment (quantitative variable) The number of animals in the compartment was determined by scoring or, if the sample size was exceeded, by information from the farm manager. The number of cubicles in the compartment was counted.	[19,20]
Median cubicle width >115 cm	Assessment by study vets	compartment	From the inner edge of one side rail to the inner edge of the opposite side rail or wall at the obvious narrowest point of the cubicle (Appendix A). Categories: yes/no	[18,21]
Median neck bar height >125 cm	Assessment by study vets	compartment	Distance from the neck bar and the lying surface (Appendix A). Categories: yes/no	[18,21]
Management of the farm	Interview with the farmer	farm	Conventional Organic or in conversion to organic farming	[20]
Grazing time	Interview with the farmer	farm	Hours/day and months/year for every group of cows was stated by the farmer. By estimating how long every cow is in the group we calculated the average time per day that a cow on this farm spends on pasture over the year: No pasture 0–2 hrs/day per animal 2–4 hrs/day per animal more than 4 h/day per animal	[20]
BCS (=body condition score)	Assessment by study vets	cow	Scoring according to the system by Edmonson (1989) modified by Metzner et al. [22] (quantitative variable)	[6,18,20]
Lameness	Assessment by study vets	cow	Scoring conducted according to a modified form of the system by Sprecher et al. [23]: 1. normal/undisturbed (=gait inconspicuous and the topline straight when standing and walking) 2. slight lameness (=slightly abnormal gait with curved back when walking) 3. moderate lameness (=abnormal gait, i.e., shortened steps with one or more limbs, with curved back line when standing) 4 + 5 (summarised) high-grade or severe lameness (=highly abnormal gait, i.e., the step is “deliberate” or there is an inability to bear weight on one or more limbs, the dorsal line is curved up when standing and walking)	[14,24]
Age of the cow	herd register	cow	<3 years 3–4 years 4–5 years >5 years	[20]
Lactation stage (=days postpartum)	DHI data	cows	Difference between date of visit and date of last calving in days (quantitative variable)	[25]

**Table 5 animals-13-02919-t005:** Prevalence of hock lesions (HL * + SHL *) at farm level in the PraeRi-Study.

Region		Hock Lesions	n	Mean	Median	STD *	Min	5% Quantile	95% Quantile	Max
HL *	North	No HL	206	27.0	18.4	24.7	0.0	5.9	44.4	94.3
		HL	206	72.0	79.8	25.0	5.7	54.7	93.0	100.0
		N/A *	206	0.9	0.0	5.3	0.0	0.0	0.0	59.2
	South	No HL	156	38.8	31.8	29.9	0.0	12.0	65.5	100.0
		HL	156	59.0	66.2	31.0	0.0	33.5	86.7	100.0
		Not assessable	156	2.2	0.0	7.4	0.0	0.0	1.3	73.2
	East	No HL	192	28.6	21.5	25.7	0.0	6.5	53.1	97.6
		HL	192	70.7	78.5	26.3	2.4	53.1	92.7	97.6
		N/A *	192	0.8	0.0	3.3	0.0	0.0	0.0	31.7
SHL *	North	No SHL	206	84.1	86.6	12.2	33.0	76.4	94.4	100.0
		SHL	206	15.0	12.5	11.3	0.0	5.5	22.6	47.6
		N/A *	206	0.9	0.0	5.3	0.0	0.0	0.0	59.2
	South	No SHL	156	85.7	89.8	14.7	26.8	76.7	97.4	100.0
		SHL	156	12.1	8.0	13.5	0.0	0.0	20.5	61.5
		N/A *	156	2.2	0.0	7.4	0.0	0.0	1.3	73.2
	East	No SHL	192	79.7	83.9	17.7	9.8	70.1	94.3	100.0
		SHL	192	19.5	14.4	17.9	0.0	5.1	29.5	90.2
		N/A *	192	0.8	0.0	3.3	0.0	0.0	0.0	31.7

* HL = hock lesion; SHL = severe hock lesion; N/A = not assessable; STD = standard deviation.

**Table 6 animals-13-02919-t006:** Hock lesions (HL)—results of the multi-factorial models in the PraeRi study.

Risk Factors	Categories	North			South			East		
		Estimated OR	CI	*p* Value	Estimated OR	CI	*p* Value	Estimated OR	CI	*p* Value
Lameness	normal gait (1)	Reference		<0.0001	Reference		<0.0001	Reference		<0.0001
	mildly lame (2)	1.2	1.1–1.4		1.5	1.3–1.7		1.4	1.3–1.6	
	moderately lame (3)	1.4	1.3–1.6		1.8	1.4–2.1		1.5	1.4–1.7	
	(severely) lame (4/5)	1.9	1.6–2.3		2.1	1.6–2.8		1.9	1.7–2.1	
Grazing time	no grazing period	1.5	0.9–2.5	0.2071	2.1	1.0–4.4	0.0589	2.3	1.0–5.0	0.2396
	>0–2 h/day per animal	1.5	0.9–2.6		1.0	0.4–2.7		2.2	1.0–5.0	
	2–4 h/day per animal	1.7	1.0–2.9		1.8	0.7–4.5		1.7	0.6–4.9	
	>4 h/day per animal	Reference			Reference			Reference		
Bedding	litter	Reference		0.1878	(A)			Reference		0.0080
	no litter or only lime	1.3	0.9–1.8					1.3	1.1–1.7	
Cubicle type	cubicles with or without a rubber mat	3.1	2.3–4.2	<0.0001	8.7	5.9–13.0	<0.0001	2.0	1.7–2.4	<0.0001
	Deep-bedded cubicles	Reference			Reference			Reference		
	cubicles with comfort mattress	2.6	1.8–3.9		9.0	4.7–17.1		5.2	3.2–8.5	
	other	2.4	1.6–3.7		(A)			0.5	0.1–2.3	
Management	conventional	4.7	1.5–14.2	0.0071	4.0	2.0–8.1	0.0001	3.0	1.2–7.9	0.0215
	organic or conversion	Reference			Reference			Reference		
Age	< 3 years	Reference		<0.0001	Reference		<0.0001	Reference		<0.0001
	3–4 years	1.3	1.1–1.4		1.3	1.1–1.6		1.3	1.2–1.4	
	4–5 years	1.8	1.6–2.0		1.8	1.4–2.2		1.7	1.5–1.8	
	> 5 years	2.4	2.2–2.7		2.3	1.9–2.8		2.5	2.3–2.7	
Median cubicle width > 115 cm	no	1.7	1.1–2.6	0.0106	1.4	1.0–2.1	0.1346	1.5	0.9–2.4	0.1425
	yes	Reference			Reference			Reference		
Median neck bar height > 125 cm	no	0.7	0.5–1.2	0.1014	(A)			(A)		
	yes	Reference								
Soiling of the lying surface	clean or single piles of faeces	(A)			(A)			Reference		<0.0001
	<50% of the surface polluted							1.4	1.2–1.6	
	>50% of the surface or completely soiled							1.0	0.8–1.2	
Body condition	increase by grade 1	0.8	0.8–0.9	<0.0001	0.7	0.6–0.8	<0.0001	0.9	0.8–1.0	<0.0001
Cow–cubicle ratio	increase by 100%	0.7	0.5–1.0	0.0267	(A)			(A)		
Lactation stage	increase by 100 days	(A)			(A)			1. 1	1.1–1.2	<0.0001

(A) = Not in model; OR = odds ratio, CI = confidence interval.

**Table 7 animals-13-02919-t007:** Severe hock changes (SHLs)–results of the multi-factorial models in the PraeRi study.

Risk Factor	Categories	North			South			East		
		Estimated OR	CI	*p*-Value	Estimated OR	CI	*p*-Value	Estimated OR	CI	*p*-Value
Lameness	normal gait (1)	Reference		<0.0001	Reference		<0.0001	Reference		<0.0001
	mildly lame (2)	1.0	0.9–1.2		1.9	1.5–2.4		1.5	1.4–1.7	
	moderately lame (3)	1.6	1.5–1.9		2.8	2.2–3.6		2.0	1.8–2.2	
	(severely) lame (4/5)	2.3	2.0–2.8		4.1	3.1–5.4		2.4	2.2–2.7	
Grazing time	no grazing period	1.5	1.1–2.1	0.0399	1.7	0.8–3.3	0.0082	0.8	0.5–1.5	0.7432
	0–2 h/day per animal	1.4	1.0–2.0		0.5	0.2–1.4		0.8	0.4–1.4	
	2–4 h/day per animal	1.2	0.9–1.7		1.2	0.5–2.9		1.1	0.5–2.4	
	>4 h/day per animal	Reference			Reference			Reference		
Bedding	litter	Reference		0.0277	Reference		0.4639	Reference		0.6477
	no litter or only lime	1.3	1.0–1.6		1.2	0.8–1.7		1.0	0.9–1.2	
Cubicle type	cubicles with or without a rubber mat	3.6	2.8–4.5	<0.0001	6.0	3.9–9.2	<0.0001	2.5	2.1–3.1	<0.0001
	deep bedded cubicles	Reference			Reference			Reference		
	cubicles with comfort mattress	3.7	2.8–5.0		6.4	3.6–11.5		3.7	2.5–5.4	
	other	2.2	1.5–3.2		(A)			1.9	0.6–5.9	
Management	conventional	2.6	1.2–5.8	0.0220	1.8	0.9–3.4	0.0902	2.6	1.3–5.3	0.0095
	organic or in conversion	Reference			Reference			Reference		
Animal age	<3 years	Reference		<0.0001	Reference		0.4338	Reference		<0.0001
	3–4 years	1.0	0.9–1.1		0.9	0.7–1.2		1.0	0.9–1.1	
	4–5 years	1.2	1.0–1.4		0.8	0.6–1.1		1.2	1.1–1.3	
	>5 years	1.3	1.1–1.5		1.0	0.7–1.2		1.3	1.7–1.4	
Body condition	increase by grade 1	0.8	0.8–0.9	0.0002	0.6	0.5–0.7	<0.0001	0.7	0.7–0.8	<0.0001
Lactation stage	increase by 100 days	(A)			(A)			1.1	1.1–1.2	<0.0001

(A) = Not in model; OR = odds ratio, CI = confidence interval.

## Data Availability

The datasets presented in this article are not readily available because the data were acquired through cooperation between different universities. Therefore, any data transfer to interested persons is not allowed without an additional formal contract. Data are available for qualified researchers who sign a contract with the project consortium. This contract will include guarantees of the obligation to maintain data confidentiality in accordance with the provisions of German data protection law. Currently, there exists no data access committee nor another body who could be contacted for the data; a committee will be founded for this purpose. This future committee will consist of the authors as well as members of the related universities. Interested cooperative partners, who are able to sign a contract as described above, may contact: MH, Clinic for Cattle at the University of Veterinary Medicine, Hannover, Bischofsholer Damm 15, 30173 Hannover, Germany, Email: martina.hoedemaker@tiho-hannover.de. Requests to access the datasets should be directed to martina.hoedemaker@tiho-hannover.de.

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
