# Peer review of "Hock Lesions in Dairy Cows in Cubicle Housing Systems in Germany: Prevalence and Risk Factors"

_animals, 2023, doi:10.3390/ani13182919_

Round 1

Reviewer 1 Report

Line 124-128. Please mention how many farms were excluded. 

Line 200-202: Please mention the number of cows that were excluded.

Line 215: Please state the number of farmers visited in your methods section clearly.  

Table 6. I am not sure if you mentioned the tools used for measuring grazing time in the methods section. 

What do think about age of cows in Table 6. Is age a confoumder? 

Paragraph one of the discussion on line 311-314 needs to be reworded. Please begin by restating what you wanted to achieve in your research before making big statements. 

Line 315 in the discussion belong to the limitations of your study. 

Line. 398: Did you want to write showed instead of sowed?

Line 419: please check on the comments above and correct appropriately. 

Line 493:.Is this a standard practice in Germany?

Line 398: You wrote sowed. Did you want to write showed?

Most of the English language is fine. 

Author Response

Dear Reviewer 1,

thank you for your helpful comments. I think your work contributed to the quality of our paper. You find the answers to your comments below (answers in green).

Line 124-128. Please mention how many farms were excluded. 

Line 200-202: Please mention the number of cows that were excluded.

Line 215: Please state the number of farmers visited in your methods section clearly.  

We added a flow chart displaying the farms and cows that were in- and excluded (Fig. 2). To our appraisal it belongs to the results section.

Table 6. I am not sure if you mentioned the tools used for measuring grazing time in the methods section. 

We state in Table 4 that the information was gained during an interview with the farmer. But maybe you are right and we should explain it in more detail. We asked the farmer which groups go to pasture for how many months a year and how many hours per day. With this information and an assumption how long a cow belongs to the groups, we calculated an average grazing time for every farm. We added this information in Table 4.

What do think about age of cows in Table 6. Is age a confoumder? 

 We think, you are right, age by itself does not promote hock lesions and is therefore also not a risk factor. Also in multifactorial analyses, age was significantly associated with hock lesions in all three regions. We see it more as a hint that the longer cows stay in the housing system, the higher the chances that they develop hock lesions as the housing conditions are not sufficient to prevent these. You find this explanation in ll .462-468.

Paragraph one of the discussion on line 311-314 needs to be reworded. Please begin by restating what you wanted to achieve in your research before making big statements. 

 We changed the paragraph, see ll. 366-372.

Line 315 in the discussion belong to the limitations of your study. 

Yes, you are right. This paragraph was meant to discuss strengths and weaknesses of our study.

Line. 398: Did you want to write showed instead of sowed?

Yes, we changed it.

Line 419: please check on the comments above and correct appropriately. 

I did not understand. Can you explain more, please?

Line 493:.Is this a standard practice in Germany?

I cannot really answer this question. The collaboration contract was set in 2016, and my impression is that in the meantime things changed. But I dont have any data on this topic.

Line 398: You wrote sowed. Did you want to write showed?

See above

Reviewer 2 Report

This manuscript summarizes a high-quality, well-conducted and well-written observational study. It aims to (i) estimate the prevalence of hock lesions on German dairy farms (cubicle housing system) in three different regions, and (ii) determine the presence of risk factors associated with severe hock lesions.

Unfortunately, I deplore the fact that this study only flies over the subject: its database is gigantic, and the possibilities immense. The authors confirm (in Germany) what is already known, published and republished. So, I would sum up the interest of the current study as confirming the high prevalence of hock lesions in dairy cows, and the associated risk factors. Its main strengths are its methodology and the number of cows sampled. Its main weaknesses are the lack of hindsight regarding the pathogenesis of the hock lesions described, and the implicit opinion in the wording of the discussion and conclusion.

Please find below my detailed comments:

Abstracts

Line 25: I understand the desire to mention the association with lameness, but perhaps keep this (very debatable) conclusion under discussion. See below for my discussion points on this subject.

Lines 35-38: Please add SD

Line 39: "higher chance" Please add at least the p-value

Line 40: No, your study is based on farms in 3 different regions, and only farms with cubicle housing systems. Please do not generalize outside your study population.

Line 41: No, you studied risk factors, not causes. Your last sentence suggests to the reader that deep-bedded cubicles are the answer to the problem. Please rephrase.

Introduction: to be honest, I think your introduction lacks a guiding thread. The data are all there, but you need to help the reader in his reasoning. On his own, he should come to the conclusion that your study is essential for improving the welfare of German dairy cows.

Lines 51-52: "However, lesions..." add reference

Lines 54-55: "A lesion is defined..." add reference

Lines 55-56: reference [5] is in Norway. You can't generalize with this single reference.

Lines 58-66: The idea is good. Unfortunately, the pathogenesis is confusing. For these 3 insults, I recommend that you start with a definition (with reference), then give an example to support your reasoning.

Line 58: "permanent"? Are you sure?

Line 60: your example of "hard surface" misleads the reader. What is the exact definition of compression (in physics)? Is compression limited to a rigid surface?

Line 61: "abrasions" or “excoriation”? See dermatology lexicon.

Lines 62-63: "If the resulting lesion..." with this second sentence you're confusing the reader. What do humidity and maceration have to do with this definition?

Line 64: Here, you don't provide a definition...

Line 67: "A common problem in diary..."  add references

Lines 68-69: "Entry of ubiquitous pathogens..." why did you add this sentence? Is this a frequent complication? Can you prove it? How many veterinarians have diagnosed septic bursitis in the field? I would delete or rephrase.

Line 80: "...with more blood..." Here you are contradicting yourself. Earlier in lines 58-60, you mentioned that compression by recumbency was responsible for a decrease in blood flow...

Line 82: here and throughout the rest of the manuscript, when quoting a published study and its associated prevalence, it is essential to define the nature of the lesions assessed, the production system and the geography. We must compare what is comparable.

Lines 84-90: You describe three distinct regions based on farming systems. Do the data collected in your study confirm this initial assumption? If so, why didn't you confirm it?

Lines 95-96: see above. What types of lesions?

Line 100: here you introduce the concept of "severe hock lesions" without having defined it beforehand...

Line 100: Why did you choose to study the risk factors of "severe hock lesions" only? What was your rationale?

Materials and Methods:

Lines 104-106: Where is your ethics committee approval?

Table 1: Where do the thresholds used in the table 1 come from?

Lines 119-123: Do "the HI-Tier” and “the association for milk testing" include all German dairy farms? Or are some farms not listed in these databases?

Line 121: "response rate" this is a result. Move to the corresponding paragraph.

Line 123: "trained study veterinarians" You don't define it. What was the training of these veterinarians? Can you verify the agreement between their assessments? If not, this is a point for discussion.

Line 124: "predominantly" Please be precise (%).

Line 126: "date of the farm visit" is the date the same for all visits? Or at least the season? If not, why didn't you take it into account in your statistical analyses? Could it create bias?

Lines 119-128: I encourage you to reformulate these paragraphs by explicitly mentioning inclusion and exclusion criteria. This will greatly help the reader. I'd also encourage you to present the results of these inclusion/exclusion criteria in the form of a flow diagram (see comments below).

Line 129: "was modified to" Why and how did you modified the score??

Line 130: "only the most severe score was..." why?

Line 131: "too dirty". Here you raise an interesting point: why is the cow dirty? Are you sure that underneath the dirt there are no skin lesions? When you exclude these cows, are you underestimating the prevalence?

Line 131: The results of these exclusions should be included in the results.

Lines 133-140: Definitions are essential here. Please define "swelling" and "wound". For the definition of "swelling", please be explicit: is it swelling in comparison to the other hock, in comparison to other cows, in comparison to what the examiner expects? Has palpation taken place? What is the expected consistency: fluctuation? Or a simple increase in volume? Is it edema, an accumulation of fluid in the bursa, in the joint? For "wound", do you distinguish between "wound", "excoriation", "pressure sore"?

Lines 131-140: are the values ordered? Is there a notion of increasing severity? If so, please explain.

Line 135: score 2 without swelling and without wound? What does "hairless area" mean? Is it simple alopecia, or callus? If you include calluses, the etiology may be different, or at least a notion of chronicity must be added.

Line 140: score 7 what are the other reasons?

Figure 1: score 4: for me, it's not a “wound” (external origin), but a “pressure sore” (internal origin). Perhaps we could talk about loss of integrity of the cutaneous barrier?

Lines 147-153: add references to find the formula used? or the software used?

Line 154: What sampling step is used?

Line 160: Table 2: I don't see your cow sampling strategy? I assume random sampling, but was it based on a list of cows on the farm? If so, I assume that the presence of an up-to-date list of cows on the farm was a inclusion criterion?

Tables 2 and 3: Can we put the information in the text to avoid overloading the manuscript with two tables?

Line 166: Were the questionnaires valid? If yes, please explain.

Table 4: Median cubicle width + median neck bar height: it's surprising that you don't adjust for cow breed and/or wither height. Please explain and/or discuss.

Table 4: "management of the farm": Not knowing the definition of "organic farm" in Germany, can we say that there are parameters specific to organic farms that are not found on conventional farms? Or is this parameter a likely confounder?

Table 4: "grazing time" Should also study the effect of season? Is humidity identical between the different seasons? Could cold influence the occurrence of certain lesions (e.g., peripheral vasoconstriction associated with cold)?

Table 4: "BCS" to be adapted to the breed?

Table 4: "Lameness" your materials and methods do not mention locomotor examination of the cows. Did you walk the cows? if so, how? did you use the same evaluators as for hock lesions? can lameness be localized to the hock? given that the majority of lameness in cattle comes from the foot, how can we look for an association between lameness score and hock lesions? Please explain or discuss.

Figures 2 and 3: of little interest... To be placed in appendix (?)

Lines 184-212: the definitions of the terms used must have been previously defined ("severe hock lesions"). See previous comments.

Lines 184-212: Here you're not comparing regions (stat), but in discussion you're alluding to them… Homogenize...

Lines 184-212: I understand the value of estimating prevalence in 3 different regions, but why separate the analysis of risk factors between the 3 regions? Shouldn't pathogenesis be the same regardless of location (region or country?)? As a reader I would have expected a statistical comparison between the 3 regions (prevalence and risk factors). What is your rationale?

Lines 202-205 : "Seven risk factors... were considered as indispensable." Please add references

Results: I recommend that you improve your manuscript by following the guidelines proposed by Animals, and/or with the STROBE Vet-Statement: e.g., when quoting a prevalence, this should always be accompanied by the numerator/denominator ratio (...% (.../...)). Whenever possible, means quoted in the text should be accompanied by SD... Statistical conclusions could repeat the (OR=...; 95% CI=...). I'd also add the flow diagram (see previous comments)

Line 221: what is the percentage of missing values? What was your strategy for dealing with missing values?

Table 5: what's the interest of all these results? What are the distributions: normal or abnormal? Can you choose to present only mean + SD or median + extremes values?

Figure 4: what's the point of this figure? aren't the results already mentioned in the other tables or in the text? moreover, such a graph loses its interest when there are no real statistical comparisons between the groups...

Figure 4: if you wish to keep this chart, I strongly suggest that you improve the table (axis titles, axis gradations, upper/lower case in the legend...).

Figure 4: Legend: "soreness" has never been defined before. I'd advise you to keep the exact same wording as the detailed score lines 134-140.

Line 255: "these results..." Delete this sentence

Lines 253-257: where necessary, complete your text with (OR=... ; 95% CI=...)

Figure 5: To delete

Line 266: "(Figure 6)" to delete

Figure 6 : to delete

Tables 6.7.8.9.10.11: the accumulation of tables confuses the reader. He doesn't know what to read or look at. Is there a way to make this more digestible? Would it be possible to put the tables in an appendix, and keep only a text containing the most important (and discussed) results? (just a suggestion.)

Discussion: I note a major effort to convey practical messages to veterinarians and breeders. Unfortunately, the wording used often reflects an implicit and unproven opinion. In the discussion, I recommend that you focus on your results, and on comparison with the pre-existing literature (with references). In the case of new hypotheses, this should be clearly explicit. All the elements to be discussed must also have been set out in the preceding paragraphs (Materials and methods, or Results). Among the elements to be discussed, I would add the limitations of your study design (see my previous comments), including sampling period, observer training, inter-observer agreement, nomenclature/lesion score used...

Line 312: "it becomes clear" implicit opinion. To rephrase.

Line 318: "free advice within the framework of the study" You don't mention it anywhere. And if this is really the case, it should have been approved by the ethics committee, since you are intervening in the health of the farm, and informing the farmer of discoveries made during a research project...

Line 319: "caused by" Have you demonstrated the causal relationship?

Lines 322-323: "enormous sample size" opinion: let the reader be the judge... Please provide precise %: 30% of all German cows? 80%? 95%?

Lines 232-234: "This made ... models" To delete

Line 325: "an internal validation was possible" what do you mean? if it's possible and recommended, I suppose you've done it? but where are these results?

Lines 327-328: are these results also in the results section? I can't find them... And apart from your surprise, what is your explanatory hypotheses? In the Material and Method section, you mention “other reasons” Can you explain them?

Line 330: "56-70%..." at farm or cow level?

Line 331: Is "problem" the right word? Your study doesn't assess the consequences of these lesions, just their presence. So, can you talk about “problems”?

Line 332: "may cause pain" also for score 1?

Line 332: "higher" where are your comparative statistics?

Line 334: "significantly higher" idem + When you add the stats, I recommend you quote the p-value when you mention a significant difference...

Line 362: "correlation" Have you made any correlation analyses?

Line 364: "preferable" opinion. To rephrase.

Line 368: "presumably higher workload" Please add reference.

Lines 364-368: Your partial conclusion is a good idea, but don't stray too far from your study. Your conclusions should be based solely on your study, not on the recommendations in the best of all possible worlds...

Line 371: "only be proven in the region East" what is your explanatory hypotheses?

Lines 378-381: see previous comments

Lines 383-391: I don't know German geography, but are pastures similar in all regions? rocky soil? relief?

Line 388: "it is conceivable..." I think your database would have answered these questions...

Lines 393-401: this paragraph should discuss the size of the cows... Line 394 "wider cubicles and higher neck bars" how can these results be interpreted without knowing the size of the cows?

Line 403: "performed significantly better" implicit opinion. To rephrase.

Lines 406-407: "organic animal husbandry... strong focus on animal welfare" is this a regulatory obligation? Can we prove it? Or is it an opinion/judgment?

Lines 411-414: Please add references

Lines 417-429: see my previous comments: are lameness frequently associated with hock lesions? your paragraph is interesting, but I'd be even more careful about the wording used. This association is debatable in the absence of an analysis of confounding factors.

Lines 430-431: "preventive... hock lesions" To delete

Lines 433-440: this hypothesis has already been proposed in the literature. Please provide references (Please note that plagiarism is not limited to copying sentences, it also applies to appropriating ideas...)

Lines 441-443: "The aim... fewer lesions." Opinion. To delete

Lines 445-449: if possible, provide references. Your paragraph is based on hypotheses, but can we say that fat accumulates easily on the lateral faces of the hocks? Is the BCS a reliable tool for assessing the amount of fat on the lateral hocks? Is the BCS perfectly correlated with the weight of the individual and the pressure exerted on the hocks during recumbency?

Conclusion: be careful not to confuse risk factors with causal factors. Please conclude only on the initial objective of your study.

Line 462: "still" did you evaluate over time or at a specific point in time?

Line 464: "central" have you assessed all possible parameters?

Line 465: " be reduced" To change to “less probable”?

Data availability statement: I'm surprised that the data can't be made public. Transparency is an explicit request of the Animals journal. Isn't it possible to present only the data presented after anonymization? We're not asking for access to all the data from the 'PraeRi' project, just those related to this manuscript.

Reviewer 3 Report

L 121: “Response rates were rather low with 9% (North and East) and 6% (South)” An explanation is given in the discussion for such a low response? Was this considered from the authors as a problem for the representative character off the study?

L 322: However, between 8,050 and 37,839 cows from 156 to 206 “farms” were examined in each region, out of how many?

L 175: Evaluation of the cubicles include, as far as concerning dimension, included only neck rail height and cubicle width. Why other crucial dimensions were not recorded? Cubicle length, brisket board length i.e. Any thought about the age of the mattresses. Especially rubber mattresses, doesn’t perform the same way as new, all lifelong.

L 398: showed instead of sowed

L 399: “Therefore, we 399 conclude that the dimensions of the cubicles are not the crucial risk factors for hock lesions” but you examined only 2 dimensions…..

L 444: Low BCS cows become lame and have more odds of hock lesions, or Cows lose BCS due to lameness and then show hock lesion? Please insist a little bit more at this point of discussion.

L 448: Reference could be altered with recent papers.

L 462: Widespread… Is it so definite? Considering the low number of farmers in the study?

Figure 4 & 5. legends all begin with capitals or not. Keep the same pace at all figures and tables.

Table 6, would be good, if it fit in one page

Table 7, check the lines

Table 8, same as 6 and 7

Table 9, check the lines

Table 10, check the lines. Split in two pages, ruins the reading

Author Response

Dear Reviewer 3,

thank you for your helpful comments. I think your work contributed to the quality of our paper. You find the answers to your comments below.

L 121: “Response rates were rather low with 9% (North and East) and 6% (South)” An explanation is given in the discussion for such a low response? Was this considered from the authors as a problem for the representative character off the study?

The explanation was given in ll. 377-380. In the database used for recruiting in northern and eastern Germany, no selection of dairy farms was possible. Therefore, the stated response rates are not totally correct, as a large proportion of contacted farms was not part of the target population as they did not meet the inclusion criterion (housing of milking cow). We conducted a non-response-analysis for small farms in northern Germany: here, 14 of 20 contacted farmers stated that they do not keep dairy cows (any more). In southern Germany, all contacted farms delivered milk and were therefore part of the target population. Here, the response rate is correct. We added this information in ll. 323-328.

We considered that the response rate probably influenced the prevalence (ll. 373 – 377). We assume that it did not bias the results of the risk factors analyses in such extent as the selection did probably not influence the relationship between risk factor and outcome. We added this information in ll. 386-387

L 322: However, between 8,050 and 37,839 cows from 156 to 206 “farms” were examined in each region, out of how many?

 We added Table 5 displaying this information.

L 175: Evaluation of the cubicles include, as far as concerning dimension, included only neck rail height and cubicle width. Why other crucial dimensions were not recorded? Cubicle length, brisket board length i.e. Any thought about the age of the mattresses. Especially rubber mattresses, doesn’t perform the same way as new, all lifelong.

 We totally agree. For these analyses, we chose only the width of the cubicles and the height of the neck rail with cut-off values as “starting point”. If we are able to make further analyses, we want to include information on cubicle length as we gained the impression that cubicle were often too short for the cows. However, there you have to split the data set into deep bedded and raised cubicles and need to adjust for the size of the cows – this was not possible during these analyses focusing on the comparison of cubicle types. Also, the interaction of width and length can be interesting… Unfortunately, we do not have the data concerning the age of mattresses but we think the performance also depends on the material and producer. It´s complex…

L 398: showed instead of sowed

 changed

L 399: “Therefore, we 399 conclude that the dimensions of the cubicles are not the crucial risk factors for hock lesions” but you examined only 2 dimensions…..

That is correct (see comment above). We changed the sentence to: “Therefore, we conclude that the width of the cubicles and the height of the neck bars are not the crucial risk factors for hock lesions“ (ll. 472-473).

L 444: Low BCS cows become lame and have more odds of hock lesions, or Cows lose BCS due to lameness and then show hock lesion? Please insist a little bit more at this point of discussion.

 The multifactorial model adjusted for lameness. Therefore, we conclude that a poor body condition (without lameness) is also associated with hock lesions. We added information that other health problems can also explain hock lesions and a poor body condition (ll. 524-525).

L 448: Reference could be altered with recent papers.

In l. 448 is no reference (at least not in our version of the manuscript).

L 462: Widespread… Is it so definite? Considering the low number of farmers in the study?

Remembering all the work, I would not consider 554 farms a low number… I do not know a study on hock lesions that included so many cows. The study of Kielland et al. included 2,335 cows from 232 farms, Barrientos et al. 76 farms, Brenninkmeyer 3,691 cows from 105 farms, Potterton et al. 3,850 cows from 77 farms, Jewell et al. 73 farms, Ekman et al. 3,217 cows from 99 farms (to be continued). To our knowledge, the study by Adams et al. (2017) comes closest to our sample size with 22,622 cows from 191 dairy farms.

However, due to the potential selection bias, we changed widespread to common.

Figure 4 & 5. legends all begin with capitals or not. Keep the same pace at all figures and tables.

 We changed Figure 4 and 5.

Table 6, would be good, if it fit in one page

 We transferred Table 6-9 to the Appendix. We did not want to change the formatting of the table. As animals is an online journal it will look different when displayed on the website.

Table 7, check the lines

 We did not find any mistakes.

Table 8, same as 6 and 7

 I do not understand. Can you explain the point?

Table 9, check the lines

Table 10, check the lines. Split in two pages, ruins the reading

 See comment above

Round 2

Reviewer 3 Report

“” L 175: Evaluation of the cubicles include, as far as concerning dimension, included only neck rail height and cubicle width. Why other crucial dimensions were not recorded? Cubicle length, brisket board length i.e. Any thought about the age of the mattresses. Especially rubber mattresses, doesn’t perform the same way as new, all lifelong.

We totally agree. For these analyses, we chose only the width of the cubicles and the height of the neck rail with cut-off values as “starting point”. If we are able to make further analyses, we want to include information on cubicle length as we gained the impression that cubicle were often too short for the cows. However, there you have to split the data set into deep bedded and raised cubicles and need to adjust for the size of the cows – this was not possible during these analyses focusing on the comparison of cubicle types. Also, the interaction of width and length can be interesting… Unfortunately, we do not have the data concerning the age of mattresses but we think the performance also depends on the material and producer. It´s complex… “”

I cannot see why, if all dimensions were included, you should “split the data set into deep bedded and raised cubicles”. You already have these categories. I think good or poor management of deep bedded cubicles should be included. Dimensions are standard when these cubicles are constructed, and only poor management spoils it. Anyway, big discussion, which needs as you say data about average cow size of each herd.

In the end, “bad” cubicles are risk factor for hock lesions.

L 476: “hypocalcaemia” There many more important disorders that diminish cow’s BCS, such as ketosis.

L 487: Reference number 36, could be altered with recent papers. By mistake I have written wrong line number in the previous review.

About the line checking the lines in some tables, it is clearly morphological. Some lines were not straight. Nothing else.

Author Response

Thank you for the explanations.

L 175: Evaluation of the cubicles include, as far as concerning dimension, included only neck rail height and cubicle width. Why other crucial dimensions were not recorded? Cubicle length, brisket board length i.e. Any thought about the age of the mattresses. Especially rubber mattresses, doesn’t perform the same way as new, all lifelong.

We totally agree. For these analyses, we chose only the width of the cubicles and the height of the neck rail with cut-off values as “starting point”. If we are able to make further analyses, we want to include information on cubicle length as we gained the impression that cubicle were often too short for the cows. However, there you have to split the data set into deep bedded and raised cubicles and need to adjust for the size of the cows – this was not possible during these analyses focusing on the comparison of cubicle types. Also, the interaction of width and length can be interesting… Unfortunately, we do not have the data concerning the age of mattresses but we think the performance also depends on the material and producer. It´s complex… “”

I cannot see why, if all dimensions were included, you should “split the data set into deep bedded and raised cubicles”. You already have these categories. I think good or poor management of deep bedded cubicles should be included. Dimensions are standard when these cubicles are constructed, and only poor management spoils it. Anyway, big discussion, which needs as you say data about average cow size of each herd.

In the end, “bad” cubicles are risk factor for hock lesions.

We totally agree.

L 476: “hypocalcaemia” There many more important disorders that diminish cow’s BCS, such as ketosis.

You are right. We picked milk fever as an example as downer cows are probably more likely to develop hock lesions. We added an "e.g."

L 487: Reference number 36, could be altered with recent papers. By mistake I have written wrong line number in the previous review.

We replaced the reference with a more recent one (also German recommendation) and also added a scientific paper.

About the line checking the lines in some tables, it is clearly morphological. Some lines were not straight. Nothing else.